# Advances in Stem Cell-Based Therapies in the Treatment of Osteoarthritis

**DOI:** 10.3390/ijms25010394

**Published:** 2023-12-28

**Authors:** Ye Chen, Rui-Juan Cheng, Yinlan Wu, Deying Huang, Yanhong Li, Yi Liu

**Affiliations:** 1Department of Rheumatology and Immunology, West China Hospital, Sichuan University, Chengdu 610041, China; chenye9735@163.com (Y.C.); crj@scu.edu.cn (R.-J.C.); wylvina@163.com (Y.W.); huangdeying99@163.com (D.H.); 2Rare Diseases Center, West China Hospital, Sichuan University, Chengdu 610041, China; 3Institute of Immunology and Inflammation, Frontiers Science Center for Disease-Related Molecular Network, West China Hospital, Chengdu 610041, China

**Keywords:** osteoarthritis, mesenchymal stem cells, therapy, extracellular vesicle

## Abstract

Osteoarthritis (OA) is a chronic, degenerative joint disease presenting a significant global health threat. While current therapeutic approaches primarily target symptom relief, their efficacy in repairing joint damage remains limited. Recent research has highlighted mesenchymal stem cells (MSCs) as potential contributors to cartilage repair, anti-inflammatory modulation, and immune regulation in OA patients. Notably, MSCs from different sources and their derivatives exhibit variations in their effectiveness in treating OA. Moreover, pretreatment and gene editing techniques of MSCs can enhance their therapeutic outcomes in OA. Additionally, the combination of novel biomaterials with MSCs has shown promise in facilitating the repair of damaged cartilage. This review summarizes recent studies on the role of MSCs in the treatment of OA, delving into their advantages and exploring potential directions for development, with the aim of providing fresh insights for future research in this critical field.

## 1. Introduction

Osteoarthritis (OA) is a chronic degenerative joint disease that commonly occurs in elderly individuals, postmenopausal women, athletes, and individuals with metabolic disorders such as diabetes and hyperlipidemia [1,2]. Its clinical manifestations include pain, joint stiffness, swelling, deformity, crepitus, and restricted mobility [3]. OA not only significantly impacts patients’ quality of life, but also stands as a leading cause of disability in the elderly [4]. With the global trend of population aging, OA is becoming a significant threat to global health.

Current approaches to OA treatment encompass physical therapy, pharmaceutical interventions, and surgical procedures. Pharmaceutical interventions primarily aim at anti-inflammatory and analgesic effects, lacking the capacity to repair or reverse joint damage associated with OA [4]. In recent years, biologic agents such as interleukin-1 (IL-1) receptor antagonists and adalimumab (TNF antibodies) have been gradually incorporated into OA treatment; however, multiple clinical trials have failed to demonstrate significant improvements [5,6,7]. Moreover, the evidence supporting the efficacy of physical therapy in OA treatment remains inconclusive [8]. Surgical interventions, such as total joint replacement and joint arthroplasty, are viable options for advanced OA patients, offering pain relief and improved joint function [9]. Nevertheless, the limited lifespan and variable long-term outcomes of artificial implants, coupled with potential complications [10], underscore the absence of definitive therapeutic modalities for joint damage repair in OA patients.

Encouragingly, recent research has unveiled the potential therapeutic role of stem cells in OA. Mesenchymal stem cells (MSCs), possessing self-renewal capabilities and derived from the mesodermal germ layer, can be harvested from bone marrow, adipose tissue, synovium, umbilical cord, dental pulp, amniotic fluid, dermis, and peripheral blood [11,12,13]. MSCs exhibit multilineage differentiation potential into osteogenic and chondrogenic lineages and secrete immunomodulatory factors, cytokines, growth factors, extracellular vesicles (EVs), and other bioactive substances, thereby contributing to tissue homeostasis and regeneration [14,15], and exerting immunomodulatory functions [16]. Numerous preclinical and clinical studies currently support the significant improvement of clinical symptoms and the repair of damaged cartilage in OA animal models and patients following MSC-based interventions [17,18,19]. In this article, we discuss the recent progress in stem cell therapy for OA, elucidating the current status of stem cell therapy as a novel approach in the treatment of OA.

## 2. Mechanisms of OA

### 2.1. Pathological Process of OA

The onset of OA is closely associated with an imbalance in cartilage metabolism, subchondral bone sclerosis, and synovial inflammation, involving various cell types, such as macrophages, T cells, chondrocytes, osteoclasts, and fibroblasts. Factors such as aging, trauma, obesity, biomechanics, and alterations in circadian rhythms can lead to hypertrophy or apoptosis of chondrocytes, metabolic disorders, cellular senescence, and disruption of cartilage homeostasis, thereby promoting the occurrence of OA [20]. In the early stages of OA, mitochondrial dysfunction in chondrocytes results in the excessive generation of reactive oxygen species (ROS), inducing oxidative damage. This process triggers chondrocyte apoptosis through the activation of the *phosphatidylinositol-3-kinase (PI3K)/protein kinase B (Akt)* and *Caspase* pathways [21,22]. Metabolic disturbances in chondrocytes, such as overexpression of glucose transporters (GLUT1), lead to increased glucose uptake and excessive production of advanced glycation end products, ultimately contributing to cartilage degradation [23]. Aberrations in lipid metabolism and cholesterol regulation also contribute to the pathogenesis of OA [24,25]. Cheng et al. observed a significant upregulation of receptor-interacting protein kinase 1 (RIP1) expression in the cartilage of OA patients and *rat* models, where RIP1, through downstream activation of bone morphogenetic protein 7 (BMP7), promoted necroptotic apoptosis in chondrocytes, contributing to OA onset [26]. Liang et al. found that downregulating RIP1 protects chondrocytes from IL-1β-induced inflammation and apoptosis [27]. Yao et al. discovered that, under inflammatory conditions, chondrocyte ferroptosis leads to increased matrix metallopeptidase 13 (MMP13) expression and downregulation of collagen II expression, contributing to OA development [28].

Various inflammatory and chemotactic factors, including colony-stimulating factors for granulocyte-macrophage (GM-CSF), IL-1, IL-6, IL-7, and IL-8, monocyte chemoattractant protein-1 (MCP-1), MCP-2, MMP1, MMP3, MMP10, and MMP13, can disrupt tissue regeneration induced by stem cells or progenitor cells, cause adjacent cell senescence, promote OA progression, and result in symptoms such as pain and impaired mobility [29].The recruitment and activation of inflammatory cells are crucial factors in the onset and development of OA. Monocyte/macrophage lineage cells, after differentiating into osteoclasts, accumulate at the subchondral plate, causing focal degeneration of subchondral bone and articular cartilage, leading to cartilage damage [30]. Activated macrophages aggregate in 76% of OA patients’ knee joints, and their quantity correlates with the severity of OA and joint symptoms such as pain, narrowed joint space, and osteophyte formation [31]. In OA animal models, Zhang et al. found that M1-polarized macrophages can secrete Rspo2 and activate β-catenin signaling in chondrocytes, exacerbating OA symptoms in *mice* model [32]. Additionally, macrophages can produce nerve growth factor (NGF), causing OA pain [33]. Macrophages secrete B-cell activating factor (BAFF), promoting proinflammatory cell responses (Th1 and Th17) and inhibiting anti-inflammatory cell responses (Th2) [34]. Synovial fibrosis in the late stages of OA is a critical factor leading to joint stiffness, synovial hyperplasia, and restricted functionality [35].

### 2.2. Mechanism of MSCs in the Treatment of OA

MSCs are multipotent stem cells with self-renewal capabilities originating from the mesoderm. They express surface markers, including CD73, CD90, and CD105, while lacking the expression of hematopoietic differentiation markers such as CD11b, CD14, CD34, CD45, CD19, CD79a, or HLA-DR [36,37]. It is essential to note species differences in MSC surface markers, with *human* MSCs commonly reported to express CD29, CD73, CD90, CD105, and CD106 [37]. *Rat* bone marrow-derived MSCs (BM-MSCs) express CD29, CD44, CD54, CD90, and CD166 [38]. Interestingly, the expression of MSC surface markers in *mice* varies slightly across strains. *C57BL, DBA1, and FVBN mice* express the SCA-1 surface marker [39], while *BALB/C mice* express SCA-1 [40]. MSCs are readily accessible and can be isolated from the stroma of almost all organs, with a relatively straight forward isolation process [41]. MSCs have demonstrated notable efficacy in the treatment of autoimmune diseases and tissue/organ repair. In the context of OA treatment, MSCs play a pivotal role in several aspects (Figure 1).

#### 2.2.1. Cartilage Regeneration

MSCs have the capability to promote cartilage formation and repair. MSCs can achieve this by differentiating into chondrocytes or by stimulating the upregulation of chondrogenic hormones in existing chondrocytes [42]. In the context of OA, MSCs can migrate from the subchondral bone to damaged areas, differentiating into chondrocytes and osteoblasts to repair cartilage and subchondral bone tissues [43]. Additionally, OA chondrocytes can stimulate the expression of type II collagen (Col2), SRY-box transcription factor 9 (SOX-9), and aggrecan in umbilical cord-derived MSCs (UC-MSCs), thereby promoting cartilage formation in UC-MSCs. MSCs can activate the *AKT* and *extracellular signal-regulated kinase (ERK)* signaling pathways in receptor chondrocytes, thereby enhancing their proliferation and cartilage formation [44]. Studies by Prasadam et al. have reported that in a coculture system of BM-MSCs and chondrocytes, the expression of cartilage regeneration markers, including Col2, aggrecan, and SOX9, is significantly higher in the BM-MSC combined with chondrocyte pellet group, confirming the chondrogenic potential of BM-MSCs [45]. Zhi et al. discovered that coculturing BM-MSCs with articular chondrocytes significantly increases the proliferation rate of chondrocytes [46]. The exosomal long noncoding RNA (lncRNA) *KLF1-AS2018* from MSCs has been found to enhance cartilage repair and chondrocyte proliferation in an OA model [47]. Furthermore, research has revealed that transforming growth factor-beta (TGF-β), as a growth factor involved in cell differentiation, proliferation, and migration, can activate the *ERK/JNK* signaling pathway, enhancing the function of vitamin D and subsequently promoting MSC proliferation and migration. This ultimately affects chondrocyte differentiation [48].

#### 2.2.2. Anti-Inflammatory and Immunomodulatory Effects

A crucial aspect of the pathogenic mechanism in OA involves the relative antagonistic effects between pro-inflammatory cytokines and anti-inflammatory cytokines. Proinflammatory cytokines such as IL-1β and TNF-α can independently or synergistically drive inflammatory responses with other cytokines [49]. Under inflammatory stimulation, damaged synovial cells or other cells release IL-1β, IL-6, IL-8, TNF-α, and MMPs with a platelet-reactive protein domain and a disintegrin and metalloproteinase with thrombospondin motifs (ADAMTS) [50]. In animal models, post-MSC treatment significantly decreases the expression of serum MMP-13, MMP-3, IL-6, and IL-8. Zhao et al. found that after MSC treatment, the levels of inflammatory biomarkers in synovial fibroblasts of OA patients, including IL-6, TNF-α, and NF-κB, were significantly reduced, while IL-10 levels increased [51].

MSCs possess specific immunomodulatory properties, producing immunoregulatory substances that downregulate immune-inflammatory processes and promote tissue regeneration, alleviating arthritis inflammation [52]. In OA treatment, MSCs inhibit innate immune responses by suppressing the development of mature dendritic cells, inhibiting IL-2-induced natural killer (NK) cells proliferation, and reducing the cytotoxicity of NK cells [53]. MSCs inhibit cell apoptosis, slow the development of T cells and B cells, and regulate adaptive immunity [54]. MSCs promote the transformation of macrophages/microglia from an inflammatory (M1) phenotype to an anti-inflammatory (M2) phenotype [55]. Additionally, MSCs inhibit the inflammatory response of NK cells by secreting TGF-β and IL-6 [56].

#### 2.2.3. Analgesic Effect of MSCs

Pain is a primary manifestation of OA and significantly impacts patients’ functionality and quality of life. In recent years, researchers have discovered that MSC therapy for OA not only promotes cartilage recovery but also significantly alleviates pain and improves function. The *cyclooxygenase 2/prostaglandin E2 (PGE2)* pathway has been implicated in the analgesic mechanism of BM-MSCs in OA [57]. Wang et al. also found that exosomes derived from MSCs modified with TGF-β1 alleviate cartilage damage and pain in OA by inhibiting angiogenesis and suppressing chondrocyte calcification and osteoclast activity [58]. Yang’s team reported the results of a clinical study involving UC-MSCs in the treatment of 44 cases of severe knee OA (KOA). The study showed that visual analog scale (VAS) scores, American Knee Society Score (AKS) joint scores, and AKS functional scores were significantly reduced at 3, 6, and 12 months after treatment, indicating that UC-MSC transplantation for severe OA can more rapidly, significantly, and persistently alleviate joint pain and improve joint function compared to hyaluronic acid sodium [59]. In a Phase I/IIa trial, late-stage KOA patients received a single intra-articular injection of 1, 10, or 50 thousand BM-MSCs. Patients demonstrated significant improvements in pain, symptoms, quality of life, and The Western Ontario and McMaster Universities Arthritis Index (WOMAC) scores overall, compared to the baseline [60]. Centeno et al. observed, through 24-weeks of follow-up magnetic resonance imaging (MRI) monitoring, that autologous transplantation of BM-MSCs can stimulate cartilage growth and reduce degenerative joint pain [61].

## 3. Application of MSCs from Different Sources in OA Therapy

MSCs sourced from different origins exhibit distinct characteristics and possess unique advantages. Bone marrow and adipose tissue are the primary sources of therapeutic MSCs [62]. Among these, UC-MSCs have the highest content [63], and MSCs derived from the umbilical cord and amniotic membrane demonstrate stronger proliferative capacity. Additionally, MSCs from the umbilical cord, amniotic membrane, and adipose tissue exhibit higher immunomodulatory capabilities than BM-MSCs, with placental MSCs having the lowest immunomodulatory strength but the ability to secrete more cell growth factors [64].

### 3.1. BM-MSCs

BM-MSCs are the most widely used source of therapeutic MSCs, offering advantages such as easy accessibility, rapid cell proliferation, strong maintenance of differentiation capabilities, and minimal risk of immune rejection [37]. Hamdalla HM et al. found that BM-MSCs significantly alleviated inflammation in a monosodium iodoacetate (MIA)-induced KOA *rat* model, improving knee joint swelling. A single intra-articular injection of BM-MSCs markedly downregulated IL-1β, IL-6, and TNF-α while upregulating IL-10 and TGF-β [41]. Chahal J et al. observed a reduction in the levels of the proinflammatory factors IL-12 and p40, decreased proinflammatory monocyte and macrophage subpopulations, and decreased synovial inflammation in late-stage KOA patients after BM-MSC injection, leading to improved passive joint mobility [60].

In a clinical trial conducted by Lamo-Espinosa JM’s team, 55 patients undergoing partial meniscectomy were recruited, and both low-dose and high-dose allogeneic BM-MSCs were used in the treatment group. A 2-year follow-up revealed no clinical adverse reactions, and based on VAS and WOMAC scores, significant pain reduction was observed [65]. Analyzing the results of two pivotal trials using autologous BM-MSCs, it was found that WOMAC responses obtained at 1-year post cell application persisted at 2 and 4 years, providing evidence for the long-term efficacy of MSC therapy for OA [66].

### 3.2. Adipose-Derived MSCs (AD-MSCs)

AD-MSCs, commonly derived from adipose tissue, possess characteristics such as abundant availability, ease of collection, minimal trauma, low complication rates, and high proliferative potential. AD-MSCs secrete anti-inflammatory factors, such as IL-10 and IL-1, inhibiting inflammation or promoting the expression of CD4+ FOXP3+ T helper cells [57].They can inhibit synovial macrophage activation, reduce the secretion of inflammatory factors, and play a regulatory role in the immune system [67]. Huňáková K et al. injected AD-MSCs into the bilateral elbow joints of *dogs* with OA and observed a decrease in the concentrations of MMP-3, TIMP-1, IL-6, and TNF-α in the joint synovial fluid [68]. Both the meta-analyses conducted by Muthu S’s team and Jeyaraman M’s team concluded that compared to BM-MSCs, AD-MSC transplantation showed better effectiveness, safety, and superiority in the treatment of KOA [69,70]. In a randomized, controlled trial aiming to assess cartilage defect regeneration, functional improvement, and safety after intra-articular injection of AD-MSCs following medial open-wedge high tibial osteotomy (MOWHTO), 26 patients were divided into an ADMSC injection group (n = 13) and a control group (n = 13). The primary outcome was the continuous change in cartilage defects assessed by periodic MRI. Continuous MRI demonstrated significantly better cartilage regeneration in the AD-MSC group than in the control group. No treatment-related adverse events, serious adverse events, or postoperative complications occurred in any case, suggesting that intra-articular injection of AD-MSCs could be a promising therapeutic approach for disease improvement after MOWHTO [71].

### 3.3. UC-MSCs

UC-MSCs possess advantages such as cellular youthfulness, large yield, high proliferative characteristics, multilineage differentiation potential, and low immunogenicity [72]. UC-MSCs can downregulate the expression of cartilage-degrading enzymes, inhibit cartilage degradation, stimulate the proliferation of damaged chondrocytes, and prevent cartilage degeneration. They also suppress pro-inflammatory cytokines such as TNF-α, IL-1β, TNF-α-stimulated protein-6, and IL-1 receptor, thereby reducing inflammation and protecting joint cartilage [73]. Ragni et al. demonstrated that the therapeutic effect of UC-MSCs in treating OA is mediated through BMP6 [74]. Zhang Q et al. found that intra-articular injection of UC-MSCs in a *rat* OA model enhanced the expression of Col2 and ki67 in joint cartilage while increasing the expression of the anti-inflammatory factors TSG-6 and IL-1 receptor antagonist, and reducing the expression of IL-1β, TNF-α, MMP13, and ADAMTS-5 in joint cartilage [73]. Research by Ju Y et al. demonstrated that the proliferation of UC-MSCs is higher than that of AD-MSCs, and both types of cells significantly reduce the development of OA induced by ACLT surgery [75]. In a Phase I pilot study conducted in 2020 on patients with bilateral KOA, intra-articular injection of UC-MSCs led to improvements in VAS, SF-36, and WOMAC scores [76].

### 3.4. Other Sources of MSCs

In addition, synovial MSCs (SMSCs), embryonic stem cell-derived MSCs (ES-MSCs), and MSCs from other sources have also been studied as potential treatments for OA. In rats with OA, injection of ES-MSCs derived from embryos showed effects such as alleviating OA pain, protecting joint structures, and slowing OA progression [77]. Akgun et al. used a combination of SMSCs (treatment group) and chondrocytes (control group) with a I/III collagen membrane to treat patients with knee joint cartilage defects. A 2-year follow-up revealed significant improvements in knee joint flexion, extension, and straight-leg raise height in both groups [78]. The mechanisms of stem cell therapy for OA vary among different cell sources. However, these approaches may be applicable to the treatment needs of different situations, and further exploration is still needed to develop precise stem cell-based therapies for OA.

## 4. Derivatives of MSCs for OA

Increasing research has found that paracrine signaling is a key mechanism through which stem cells exert their effects [79]. MSCs can secrete various growth factors and nutrients, promoting their own differentiation into chondrocytes, stimulating the proliferation and growth of resident chondrocytes, and secreting anti-inflammatory and immunomodulatory factors to regulate the damaged tissue microenvironment and promote tissue regeneration [80,81]. The paracrine effects of MSCs are mediated by the secretion of EVs [82]. EVs encompass various membrane-bound vesicle structures released by cells. According to their size and biogenesis, EVs can be classified into three subtypes: exosomes, microvesicles, and apoptotic bodies [83]. Exosomes are one subtype, characterized as nanoscale (40–100 nm) vesicles spontaneously produced and secreted by various active cells in mammals [84].

Compared to MSCs, EVs retain similar biological characteristics but offer advantages such as biocompatibility, low immunogenicity, long-term stability, and ease of storage [85,86]. EVs derived from BM-MSCs can alleviate joint inflammation by inhibiting the NF-κB pathway [87]. Those from AD-MSCs can protect cartilage and alleviate inflammation by inhibiting the *PI3K/Akt/mTOR* pathway [88]. Studies have found that exosomal miR-92a-3p can promote the proliferation and differentiation of chondrocytes, reduce cartilage degradation, and activate MMPs, thereby improving OA [89]. Exosomal miRNAs can mediate cell-to-cell communication and gene regulation, including the regulation of cartilage formation and degeneration, making them ideal substitutes for cell-free MSC therapies for OA [90]. In addition to their ability to regenerate cartilage, EVs can prolong the survival of degenerative chondrocytes induced by IL-1β by increasing the expression of chondrocyte markers and simultaneously inhibiting the expression of degradation-related genes (including ADAMTS-5 and MMP-13). This helps in upregulating Col2 and resynthesizing the extracellular matrix (ECM), protecting cartilage, preventing chondrocyte hypertrophy, and avoiding chondrocyte dedifferentiation [91]. The powerful ability of EVs derived from MSCs to improve and protect joints from injury-induced degeneration has been confirmed in preclinical studies. For example, studies by Cosenza et al. demonstrated that EVs and exosomes derived from MSCs could enhance the expression of chondrocyte biomarkers (Col2 and aggrecan), reduce inflammatory mediators, protect chondrocytes from apoptosis, inhibit macrophage activity, and emphasize the equivalent protective role of EVs and exosomes in the pathogenesis of OA [92]. Research by Li K et al. indicated that EVs from UC-MSCs could provide therapeutically effective miRNAs. These miRNAs could promote the reprogramming of macrophages into the M2 type, induce the expression of IL-10, and effectively induce the *PI3K-Akt* signaling pathway, which could delay knee joint cartilage degeneration in OA [93]. Another characteristic of OA progression is chondrocyte apoptosis. Studies by Jin et al. found that BM-MSC-exosomes could reduce IL-1β levels through lncRNA *MEG-3*, alleviate chondrocyte apoptosis and senescence, and have the potential for cartilage protection and improvement of osteoporosis [94]. *MiRNA-140* is specifically expressed and regulates ECM-degrading enzymes in cartilage. Research by Si HB’s team concluded that intra-articular injection of *miRNA-140* could alleviate the progression of OA by regulating the extracellular homeostasis of *rats* [95]. As a carrier for delivering *miR-140* to chondrocytes, exosomes have become a new treatment for OA. This targeted treatment, using exosomes as drug carriers, not only improves the efficacy of anti-OA but also reduces the cost of treatment for patients [96].

MSC-ECM is also a noncellular component that contains various large molecules secreted by MSCs. After removing cell components that trigger immune reactions such as DNA, the ECM retains natural biochemical and biophysical signals [97]. Platas J et al. found that conditioned medium from AD-MSCs could downregulate the production of IL-1β in OA chondrocytes [98].

Another noteworthy cell therapy is bone marrow aspirate concentrate (BMAC). This therapy stands out for its composition, which includes multiple cell fractions such as platelets, monocytes, and MSCs [99]. Clinically, BMAC is increasingly utilized as a regenerative therapy for musculoskeletal pathological conditions, although its evidence-based support remains limited. Multiple clinical trials have indicated the effectiveness of BMAC therapy in reducing OARSI intermittent and persistent joint pain, as well as bilateral knee VAS pain scores. Moreover, it has shown improvement in pain and patient-reported outcomes in KOA patients during short-term follow-up [100,101,102,103]. Additionally, a study followed up with 140 patients for 15 years and found that subchondral BMAC had a significant effect on pain, leading to the postponement or avoidance of total knee arthroplasty in the contralateral joint of patients [104]. Consequently, intra-articular autologous BMAC therapy has been deemed safe, effective in managing pain, and capable of ameliorating functionality in patients with symptomatic KOA [104,105,106].

## 5. Application of Novel Techniques for MSCs

### 5.1. Preconditioning

MSCs exhibit dynamic adaptability to their environment, adjusting their regulatory functions based on different conditions. To enhance MSC functions both in vitro and in vivo, essential strategies include preconditioning, genetic modification, and optimizing MSC culture conditions. The implementation of these procedures holds the promise of significantly improving the efficacy of MSC transplantation in tissue engineering and regenerative medicine [107].

During the tissue regeneration process, MSCs often encounter challenges such as low cell viability, poor differentiation, ineffective cell communication, suboptimal ECM formation, and slow migration [108]. Preconditioning of MSCs has been employed to enhance their capabilities in vascular induction, proliferation, survival, immunomodulation, and migration toward injured areas [109].

MSCs subjected to hypoxic preconditioning can secrete increased quantities of bioactive factors, including vascular endothelial growth factor (VEGF), hepatocyte growth factor (HGF), insulin-like growth factor-1 (IGF-1), and basic fibroblast growth factor (bFGF), leading to enhanced angiogenesis and osteogenic induction [110]. TGF-β1 stimulation induces CD24 expression in MSCs while concurrently reducing surface expression of MHC I and MHC II [111,112]. Wan Safwani WKZ and colleagues found that hypoxia improves the viability and proliferation rates of cryopreserved AD-MSCs. This process upregulates hypoxia-inducible factor 1 alpha (HIF-1α) and induces the expression of chondrogenic genes such as aggrecan, Col-2, and SOX-9 in AD-MSCs, promoting their proliferation, differentiation into chondrocytes, and facilitating cartilage repair [113].

Chemical or pharmaceutical preconditioning is another strategy employed. *Toll-like receptor (TLR)* activation plays a crucial role in immune responses and MSC differentiation. For instance, stimulation with LPS, a TLR4 agonist, does not impact the viability or morphology of BM-MSCs, nor does it alter the expression pattern of key cell surface markers. However, it does influence the immunomodulatory potential of MSCs [114,115]. Preconditioning with LPS improves the regenerative properties of MSCs, significantly promoting the proliferation and migration of chondrocytes while inhibiting chondrocyte apoptosis [116]. Research indicates that pretreatment with a *p38/MAPK* inhibitor (sb203580) leads to increased ECM expansion in the inflammatory environment, enhancing the ability of *human* SMSCs to regenerate cartilage [117]. Preconditioning with IL-3 promotes the upregulation of chemokine receptor 4 (CXCR4) expression in MSCs, increasing their migration and facilitating their differentiation into osteogenic cells [118].

### 5.2. Genetic Engineering

The utilization of gene editing techniques to knockout or silence genes inhibiting the osteogenic or chondrogenic differentiation of stem cells and the transfection of genes favoring osteogenic or chondrogenic differentiation has emerged as a novel research approach in the field of stem cell regenerative medicine. Transfection of *human* UC-MSCs induced with a recombinant *pIRES2-EG-FP-hBMP-2* plasmid showed stable transcription of the *hBMP-2* gene, indicating successful differentiation of stem cells into chondrocytic tissues [119]. Under conditions of hyperglycemia, the knockdown of *NLRP3* demonstrates a protective effect on the paracrine function of AD-MSCs. This protection is achieved through the activation of anti-apoptotic and anti-ROS deposition mechanisms [120]. Studies have revealed that genetic synthesis modifications of certain miRNAs in MSC-exosomes can result in improved cartilage repair outcomes [47,121]. MAO et al. found that overexpression of *miR-92a-3p* in exosomes derived from BM-MSCs exhibited enhanced inhibition of early-stage OA progression in a collagen-induced mouse model [89].

### 5.3. MSC Transplantation and Novel Technologies

The combination of MSCs with biomaterials has demonstrated notable efficacy in the treatment of OA [122]. The key feature lies in the interaction between biomaterials and MSCs, guiding the differentiation of MSCs into specific cell lineages. Their combined application in OA models has been shown to promote hyaline-like cartilage regeneration, opening new avenues for diverse clinical applications of MSCs [123].

Biomaterials possess precise shapes and microstructures, exhibit stable performance, are nontoxic, and display excellent biocompatibility. When implanted into cartilage-deficient areas along with MSCs, these biomaterials degrade, aiding in the chondrogenic differentiation of MSCs and facilitating the formation of new cartilage [124]. Commonly used scaffold biomaterials include collagen gel, agarose, platelet-rich fibrin gel, and 3D scaffolds [125,126]. KIM et al. investigated the role of self-assembling peptide hydrogelen capsulated MSCs in treating *rat* OA models, demonstrating chondroprotective effects, suppressing inflammatory cytokine levels and apoptosis biomarker expression, along with reducing subchondral bone density [127]. Using a hyaluronic acid gel sponge as a carrier, Kayakabe M et al. effectively transplanted BM-MSCs into *rabbit* joints, leading to the successful repair of damaged articular cartilage [128]. In the first large-scale clinical trial conducted by WAKITANI et al., autologous BM-MSCs embedded in a collagen gel were implanted into the cartilage-deficient areas of OA patients, resulting in improved joint function in both the control and BM-MSC groups [129]. Haleem et al. suggested that intra-articular implantation of a fibrous gel cell scaffold containing autologous BM-MSCs and platelet-rich plasma might be a more effective treatment for repairing joint cartilage injuries [130]

## 6. Conclusions

MSCs have been extensively researched since entering the scientific spotlight, with their role in various diseases being widely explored. The positive impact of MSCs on OA treatment has been substantiated through clinical experiments. Several controlled, randomized Phase I/II trials have demonstrated that patients treated with UC-MSCs and BM-MSCs experienced significant improvements in pain and function, compared to the HA-treated group at 12 months from the baseline. Moreover, almost no adverse effects were reported after MSC administration or during short-term follow-up [105,131,132]. While the majority of studies support the positive effects of MSCs therapy in OA treatment, a randomized Phase 3 trial has shown no significant advantage of BM-MSC over corticosteroids in terms of reducing pain scores and improving MRI scores [133]. Moreover, EVs derived from MSCs have demonstrated promising therapeutic effects in multiple experiments involving OA animal models. While significant efforts are needed for advancements in both basic and clinical research frontiers, improved cell therapies hold the promise of becoming safer, more affordable, and more effective treatments for OA. The pretreatment of MSCs, genetic engineering enhancements, and their combination with biomaterials further amplify the advantages of MSC-based treatments for OA, providing additional avenues for the treatment of OA and potentially other diseases. In the future, we anticipate broader recognition and application of MSCs in the field of OA and beyond.

## Figures and Tables

**Figure 1 ijms-25-00394-f001:**
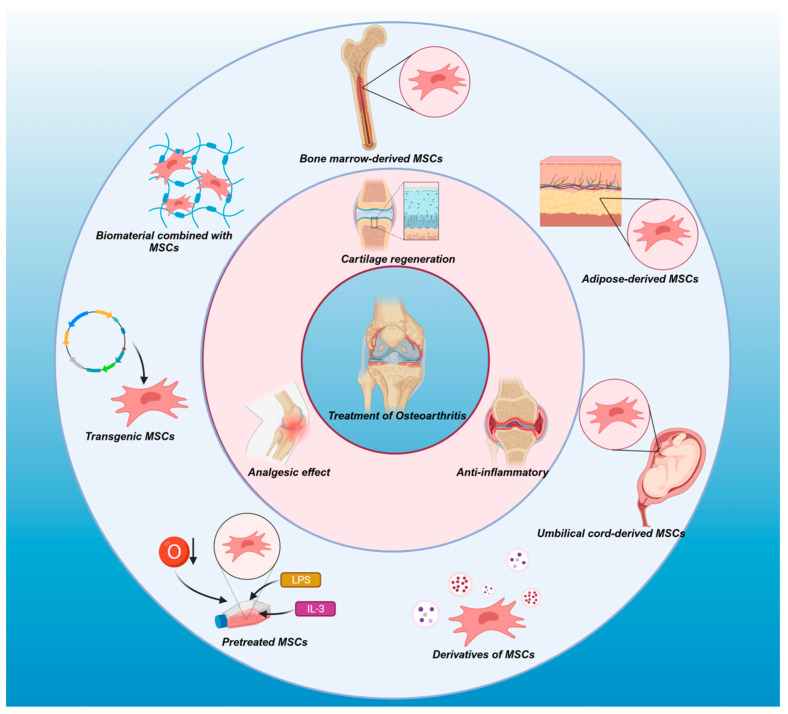
Different sources of mesenchymal stem cells (MSCs)—including derivatives, pretreated MSCs, and transgenic MSCs, as well as biomaterials combined with MSCs—contribute to cartilage regeneration, anti-inflammatory response, immune regulation, and analgesic effects. These versatile applications position MSCs as potential therapeutic agents for osteoarthritis (OA), offering the prospect of slowing down the progression of OA.

## Data Availability

Not applicable.

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
