# Peer review of "Advances in Stem Cell-Based Therapies in the Treatment of Osteoarthritis"

_ijms, 2023, doi:10.3390/ijms25010394_

Round 1

Reviewer 1 Report

Comments and Suggestions for Authors

It seems to me to be an excellent review work of MSCs on OA and very well structured and based on a vast, current and adequate bibliography. This review addresses three important research questions:

 The mechanisms of mesenchymal stem cells in the treatment of osteoarthritis;

The sources of MSC for OA therapy;

Derivates of MSCs.

We consider the topic original and relevant in the field of OA, as there is still no ideal therapy. Compared with other published material seems to me an excellent review.

So, I only suggest minor changes.

MSCs don`t need to put in full in all work sections

Figure 1 Legend: Put MSCs and OA in full; change “transgenis” to “transgenic”

Author Response

Comments:

MSCs don`t need to put in full in all work sections. Figure 1 Legend: Put MSCs and OA in full; change “transgenis” to “transgenic”.

Reply:

Thank you very much for your high praise of our review. We have revised the manuscript according to your constructive comments. We have changed all terms to abbreviated forms, except for first occurrences, such as osteoarthritis, mesenchymal stem cells, and extracellular vesicles. We have also corrected the legend of Figure 1.

Reviewer 2 Report

Comments and Suggestions for Authors

Thank you for the opportunity to review this well written manuscript summarizing MSC therapy to treat osteoarthritis. See few comments below. Well done!

Consider integrating some further discussion of terminology used (stromal versus stem cells) and appropriateness of cell surface markers to define MSC as well as potential species differences in doing so.

Recommend expansion of literature regarding up or down regulation of specific cell surface markers in the presence of inflammation or pretreatment (TGFB, TLR, NLR agonists) and discussion of effects of preconditioning agents in general.

Recommend some mention of bone marrow aspirate concentrate in treatment of OA (recognizing it has a low MSC content), perhaps under derivatives section.

Author Response

Comments:

Consider integrating some further discussion of terminology used (stromal versus stem cells) and appropriateness of cell surface markers to define MSC as well as potential species differences in doing so.

Recommend expansion of literature regarding up or down regulation of specific cell surface markers in the presence of inflammation or pretreatment (TGFB, TLR, NLR agonists) and discussion of effects of preconditioning agents in general.

Recommend some mention of bone marrow aspirate concentrate in treatment of OA (recognizing it has a low MSC content), perhaps under derivatives section.

Reply:

Thank you very much for your high praise of our review.We appreciate your constructive comments. We conducted a thorough search of the pertinent literature and incorporated the corresponding content into the review at the designated locations, as elaborated on page 3, page 8 and page 9.

Reviewer 3 Report

Comments and Suggestions for Authors

The work by Chen et al. is an interesting review of the use of MSCs-based therapies in the treatment of osteoarthritis. The review addresses the topic by analyzing the aspects of the biology of MSCs that can contribute to the therapeutic effect by presenting data on clinical applications in humans or animal models.

The review is well-written and easily understandable. However, I highlight that several citations (for example, 17, 58, 60) need to be corrected, and therefore, I suggest carefully reviewing the entire bibliography and its numbering in the text. The correctness of the references is an essential aspect.

A further comment on the work concerns the fact that, despite citing a high number of bibliographical references relating to the clinical application of MSCs in osteoarthritis, no works are reported that in some way highlight aspects that still need to be clarified, not so much on safety, but above all on the clinical efficacy of treatment with OA cells. As an example, I give the recent work by Mautner et al. ( Nat Med (2023). https://doi.org/10.1038/s41591-023-02632-w), which discusses the comparison between cells and corticosteroids in a large clinical trial in humans. A complete review should also include these types of results, providing a complete picture of the knowledge relating to the clinical efficacy of cell-based treatments in OA and, more generally, the clinical effects of MSCs. I suggest the authors evaluate the introduction of such papers.

Comments on the Quality of English Language

The language is fine, and the manuscript is easily readable. 

Author Response

Comments:

Reviewing the entire bibliography and its numbering in the text. The correctness of the references is an essential aspect.

A further comment on the work concerns the fact that, despite citing a high number of bibliographical references relating to the clinical application of MSCs in osteoarthritis, no works are reported that in some way highlight aspects that still need to be clarified, not so much on safety, but above all on the clinical efficacy of treatment with OA cells. As an example, I give the recent work by Mautner et al. ( Nat Med (2023). https://doi.org/10.1038/s41591-023-02632-w), which discusses the comparison between cells and corticosteroids in a large clinical trial in humans. A complete review should also include these types of results, providing a complete picture of the knowledge relating to the clinical efficacy of cell-based treatments in OA and, more generally, the clinical effects of MSCs. I suggest the authors evaluate the introduction of such papers.

Reply:

Thank you very much for acknowledging our review and the valuable suggestions provided. We have meticulously re-evaluated the placement of all references in this review and made necessary corrections. Furthermore, we conducted an additional literature search to include studies evaluating the effectiveness of MSCs for the treatment of osteoarthritis.